# The Current Findings on the Impact of Prenatal BPA Exposure on Metabolic Parameters: In Vivo and Epidemiological Evidence

**DOI:** 10.3390/nu14132766

**Published:** 2022-07-05

**Authors:** Hala F. S. Abulehia, Noor Shafina Mohd Nor, Siti Hamimah Sheikh Abdul Kadir

**Affiliations:** 1Institute of Medical Molecular Biotechnology, Faculty of Medicine, Universiti Teknologi MARA (UiTM), Cawangan Selangor, Kampus Sungai Buloh, Jalan Hospital, Sungai Buloh 47000, Selangor, Malaysia; halalehia@gmail.com; 2Department of Paediatrics, Faculty of Medicine, Universiti Teknologi MARA (UiTM), Cawangan Selangor, Kampus Sungai Buloh, Jalan Hospital, Sungai Buloh 47000, Selangor, Malaysia; 3Institute for Pathology, Laboratory and Forensic Medicine (I-PPerForM), Faculty of Medicine, Universiti Teknologi MARA (UiTM), Cawangan Selangor, Kampus Sungai Buloh, Jalan Hospital, Sungai Buloh 47000, Selangor, Malaysia; sitih587@uitm.edu.my; 4Department of Biochemistry and Molecular Medicine, Faculty of Medicine, Universiti Teknologi MARA (UiTM), Cawangan Selangor, Kampus Sungai Buloh, Jalan Hospital, Sungai Buloh 47000, Selangor, Malaysia

**Keywords:** bisphenol A, endocrine disruption, metabolic disorder, diabetes mellitus, obesity, insulin resistance

## Abstract

Metabolic syndrome (MS) is a multifactorial disease entity and is not fully understood. Growing evidence suggests that early exposure to bisphenol A (BPA) is a significant risk factor for the development of metabolic diseases. BPA is a monomer used in the manufacturing of polycarbonate plastics, thermal receipt paper, and epoxy resins. Owing to its widespread use, BPA has been detected in human fluids and tissues, including blood, placental breast milk, and follicular fluid. In the present review, we aimed to review the impact of prenatal exposure to different doses of BPA on metabolic parameters as determined by in vivo and epidemiological studies. The PubMed, Scopus, and Web of Science electronic databases were searched to identify articles published during a period of 15 years from 2006 to 2021, and 29 studies met the criteria. Most studies demonstrated that prenatal exposure to low BPA concentrations correlated with alterations in metabolic parameters in childhood and an increased risk of metabolic diseases, such as obesity and type 2 diabetes mellitus (T2DM), in adulthood. Therefore, prenatal exposure to low doses of BPA may be associated with an increased risk of obesity and T2DM in a sex-specific manner.

## 1. Introduction

Bisphenol A (BPA) has been industrialised and is widely used in various applications. BPA is a key unit (monomer) in the manufacturing of polycarbonate plastics, thermal receipt paper, and epoxy resins. These materials are used in food containers and beverage bottles, including the linings of food cans [1]. BPA can also be found in sealants used in various applications, including medical equipment and in dentistry [2]. Owing to its pervasive use, BPA has become a ubiquitous pollutant in food, beverages, wastewater, air, dust, and soil [3]. Therefore, the consumption of food and drinks can be a major source of BPA exposure [4]. Many studies have reported the detection of BPA in human fluids and tissues, including blood, pregnancy-associated fluids (placental and amniotic fluids), breast milk, follicular fluid, umbilical cord blood, urine, and adipose tissue [5,6,7]. When BPA products are exposed to heat or an acidic environment, the BPA monomer readily leaches from the epoxy resin that lines cans, food packaging, and thermal paper receipts. This leads to increased human exposure through the skin and mouth or via the placenta and breast milk from mother to offspring [8,9,10]. BPA is predominantly found in maternal and foetal matrices; thus, BPA is easily transferred through the placental membrane to the foetus [11].

BPA is a well-known endocrine disruptor that has obesogenic effects. According to epidemiological, in vitro, and in vivo studies, BPA has many effects on human health, such as cardiovascular diseases [12,13], obesity [14], T2DM [15,16], decreasing neurogenesis, negatively impacting the brain and behaviour, impairing learning, contributing to memory performance impairment [17,18], and affecting fertility [19]. A large body of evidence links BPA to adverse health effects on metabolism through epidemiological, in vivo, and in vitro studies. Although many studies have supported the assertion that environmental exposure to BPA may be detrimental to human health, particularly due to its link with metabolic syndrome (MS), to the best of our knowledge, there has been no review comparing and contrasting the findings on BPA exposure with the metabolic parameters reported by many researchers. MS is an asymptomatic pathophysiological condition characterised by obesity, insulin resistance (IR), dysglycaemia, hypertension, and dyslipidaemia. In general, MS includes a large waist circumference, high triglyceride (TG) level, low high-density lipoprotein (HDL) cholesterol level, increased blood pressure, IR, and elevated fasting blood glucose [20].

Previous reviews have discussed the relationship between prenatal BPA exposure and its effects on a wide variety of metabolic disorders, such as obesity and T2DM [21,22,23,24,25]. However, when we scrutinised the outcomes of the studies, we discovered a controversy regarding the effect of BPA in the findings of the studies. While some studies found that there was a relationship between BPA exposure and MS [3,26], others showed the contrary [27], while others called for more investigation into BPA’s effects as an environmental obesogenic [14].

These differences between the studies may be attributed to the exposure windows and BPA doses, which varied between studies. In this study, we reviewed the effects of different doses of prenatal BPA exposure on metabolic parameters as determined by in vivo and epidemiological studies. The window of exposure is critical and serves as a determinant of the magnitude and permanence of the adverse outcomes of BPA exposure. The existence of “critical windows of development” during which developing systems are especially sensitive to hormonal or other disturbances means that exposure to BPA prenatally or early in development can have long-term consequences [5,9]. Many human studies have identified links between maternal BPA exposure during pregnancy and offspring outcomes. Therefore, we highlight the prenatal period, which is a critical disease development period according to studies, specifically for metabolic diseases, including obesity, T2DM, and cardiovascular diseases [12,28,29].

The controversy regarding the effect of BPA that is evident in the varying results of the studies may be attributed to the wide range of dosages used in different study designs. To investigate the dose effects of BPA exposure, this review divided exposure into three groups: very low, low, and high dose levels. Based on previous reports, a very low dose was defined as a dose lower than the tolerable daily intake (TDI) (4 µg/kg/day), while low doses (4–50 µg/kg/day) and high doses of BPA were set higher than the LOAEL (50 µg/kg/day) for animal studies.

In this review, we investigate the effect of prenatal BPA exposure at different doses on metabolic parameters, which are crucial for the prognosis and diagnosis of metabolic diseases. To the best of our knowledge, there has been no review of the relationship between BPA exposure and metabolic parameters in children, particularly with respect to in utero exposure to BPA.

## 2. Search Strategy

A literature search was conducted using the Scopus, Web of Science, and PubMed electronic databases, for a 15-year-period from 2006 to 2021. The search was conducted using the Boolean operators “AND” and “OR”: “bisphenol-A OR BPA” AND “perinatal” AND “metabolism” AND “insulin resistance” AND “glucose” AND “lipid” (we used general terms to ensure that we collected all articles that included the biochemical parameters). The search was limited to English-language publications. The titles and abstracts of the studies were screened. The full texts of potentially relevant articles were retrieved. An illustration of this process is shown in Figure 1.

The inclusion criteria for this review were prenatal BPA exposure in children and in vivo models with disrupted metabolism homeostasis. Since the aim of this review was to identify the impact of prenatal BPA exposure on metabolic parameters in children, articles that did not include a report on metabolic parameters were excluded. An overview of the inclusion and exclusion criteria is presented in Table 1. Interventionary studies involving animals or humans, and other studies that required ethical approval, were required to list the authority that provided approval and the corresponding ethical approval code.

## 3. Sources of BPA Exposure and BPA Metabolism

BPA is widely used in the manufacturing of many consumer products. This results in consumer dietary exposure to BPA. Thus, humans can easily be exposed, including mothers and infants. However, BPA exposure in humans can also occur via water, air, and soil. Nevertheless, the majority of exposure comes from the consumption of BPA-contaminated foods and beverages. Food, particularly canned food, is often regarded as the most significant source of BPA. Food contamination with BPA is typically caused by contact with food packaging products containing epoxy resins and polycarbonate monomers [29].

The predominant source of BPA is polycarbonate, which includes food-contact materials such as baby bottles, food containers, and epoxy resins that are used as cover linings for canned beverages and food [30,31]. Dietary and non-dietary exposure to BPA sources is summarised in Figure 2. Based on the available data from previous studies, it is apparent that exposure to BPA from dietary sources is higher than that from non-dietary items. Food was shown to contribute to more than 90% of the total exposure to BPA, while exposure from other sources accounted for less than 5% for all age groups [30].

BPA was detected in human serum in low concentrations ranging from 0.0002 to 66 ng/mL throughout the general population. Furthermore, BPA levels in umbilical cord blood averaged from 0.5 to 52.26 lg/L, indicating both maternal and foetal exposure. BPA is rapidly absorbed when taken orally, reaching its maximum blood concentration in 2 h with a half-life of approximately 6 h [14].

Generally, an adult’s liver is able to eliminate BPA from the body. Over 90% of BPA is metabolised primarily into an inactive metabolite, BPA-glucuronide (BPA-GA), which is excreted mainly in the bile [14,32]. However, studies have reported a decrease in the glucuronidation and excretion rates in pregnant rats and an increase in BPA-GA in their blood, thus increasing the risk of the transfer of BPA-GA to foetuses via the placenta. In addition, BPA-GA is deconjugated into BPA (an active metabolite) in foetuses [32]. Moreover, various experiments have shown that BPA can immediately pass through the placenta to foetuses [33,34]. Foetuses have immature livers and a weak drug-metabolising system, which could increase the adverse effects of BPA on foetuses [32].

## 4. Different Doses of BPA Exposure

The detection and toxicological analysis of various chemicals are critical for understanding environmental pollutants and their risks, as well as how these pollutants may affect public health at multiple levels. Practical experimental research is crucial for discovering the lowest doses that pose a health risk, which is known as the lowest observed effect level (LOAEL), and the doses that do not pose a health risk, known as the no observed adverse effect level.

Despite the fact that a large amount of research on the toxic effects and hormonal activity of BPA in animal models has been published, there have been significant differences in the outcomes of these studies in terms of both the nature of the effects observed and the levels at which they occur. Several studies have demonstrated that exposure to BPA, even at low doses, can result in adverse health effects. Infants, children, and pregnant women are those most harmed by exposure to BPA. There is inconsistency in the results of studies on whether in utero exposure to BPA leads to the remodelling and alteration of the metabolisms of children. There is also controversy about the dose that causes defects in metabolic parameters in children. According to the U.S. Environmental Protection Agency (EPA), the reference dose for tolerable daily BPA exposure for the human population without any considerable risk of deleterious effects during life is 50 µg/kg/d, according to rodent studies. In 2015, the European Food Safety Authority reduced the toxicological threshold for BPA from 50 to 4 µg/kg/d [35].

## 5. The Mechanisms of Action of BPA and Its Deleterious Effects

BPA is a lipophilic component that is considered an endocrine disruptor compound and acts as an oestradiol hormone. In vitro and in vivo studies have shown that BPA can bind to various nuclear receptors, such as receptors ERα and β, G protein-coupled receptor 30 (GPR30), androgen receptor (AR), thyroid hormone receptors α and β, oestrogen-related receptor gamma, and glucocorticoid receptor [36]. Oestrogens regulate a variety of physiological functions, such as development, growth, and tissue homeostasis, by binding to and activating the oestrogen receptors ERα and ERβ. Although the affinity of BPA for the oestrogen receptor is weaker and its activity is roughly 10,000 to 100,000 times lower than that of the natural hormone 17 beta-oestradiol(E2), BPA nevertheless reacts like oestradiol, triggering various cell responses. The BPA compound interacts with ERα through 42 van der Waals bonds instead of the 51 bonds involved in the E2-ERα interaction [37].

In vitro experiments have shown that the effect of BPA on specific tissues is similar to that of oestrogen [37]. BPA can inhibit adiponectin (ADP) secretion and stimulate the secretion of inflammatory adipokines, such as tumour necrosis factor α and interleukin-6, from human adipose tissue [38]. BPA may also affect body weight (BW). One study described alterations in BW dependent on sex and dose potential in CD-1 mice after developmental exposure to BPA [39]. Recently, in addition to the effect of BPA on nuclear receptor genomic signalling, it has been suggested that the deleterious effects of low-dose BPA on human health may be mediated by membrane receptors in a non-genomic manner to trigger rapid biological reactions on specific cellular targets. The signalling pathway, including GPR30, a non-classical ER, has been identified as a seven-transmembrane domain receptor and is highly significant in the adverse effects of low-dose BPA [36]. An in vitro study by Wang et al. showed that knockout (GPRKO) GPR30 protected female mice from obesity induced by a high-fat diet, IR, and blood glucose intolerance [40]. Interestingly, a study showed that low doses of BPA (0.1 nM) increased the expression of inflammatory molecules and induced GPR30 in cultured mature adipose tissue and in stromal vascular fraction cells isolated from mammary human adipose tissue biopsies [41]. In addition, BPA has an anti-androgen effect; thus, it can compete with 5α-dihydrotestosterone for binding to ARs. Several studies have shown that BPA can bind to AR through hydrophobic interactions [36]. Interestingly, sex-dependent serum and urine BPA concentrations have been observed, possibly due to the variation in androgen-related enzyme activity between males and females. Exposed males (both foetuses and adults), obese women, and women with polycystic ovary syndrome showed higher levels of serum BPA compared with healthy females exposed at the same period and in the same doses [42].

In summary, BPA has deleterious effects with respect to the cardiovascular system, cancer, metabolism, immunity, and the reproductive system by binding to a specific nuclear receptor, including ERs and ARs, as well as thyroid hormone receptors; by recruiting transcription factors; by regulating epigenetic modifications; and through non-genomic mechanisms using membrane receptors, including GPR30.

## 6. Discussion

### 6.1. Metabolic Effect of Prenatal BPA Exposure in Animal Studies

#### 6.1.1. High and Low Doses of Prenatal BPA Exposure and Glucose Parameters in the Animal Models

Several studies have shown that prenatal BPA exposure affects pancreatic β-cells, insulin secretion, and glucose metabolism. Some also suggest that BPA has an obesogenic and diabetogenic action, especially pronounced when exposure occurs in the early stage of development in humans and animals, which is a critical period of growth and differentiation of metabolically active tissues. It was confirmed by different experiments that BPA induced metabolic disorders in human and animal models of prenatal exposure (Table 2).

Long et al. suggested that gestational exposure from gestational day (GD) 7.5 until GD 16.5 to a low BPA dose (1 µg/kg/day) in C57BL/6J mice was associated with sex-dependent glucose and lipid metabolic dysfunction. The study’s findings showed that exposure in adult males (14 weeks), but not females, to low doses of BPA caused increased hepatic TG and glycogen levels, in addition to significantly increased levels of fasting blood glucose, insulin, IPGTT, and IPITT. Interestingly, BPA levels were found to be increased in the serum of male mice exposed to 1 μg/kg/day BPA, but no significant difference was observed in female mice relative to non-BPA-exposed (control) mice [43]. Another study conducted by Diamnte et al., (2021) did not show any effect on plasma insulin levels or glucose tolerance in female offspring at 10 or 21 weeks. Male exposure to low gestational BPA significantly decreased BW at 4 weeks and induced a faster glucose clearance based on IPGTT results at 10 and 21 weeks [44].

The study was conducted to investigate developmental exposure to a very low dose of BPA; interestingly, in Fischer 344 rat offspring, it was demonstrated that exposure to a very low BPA dose (0.5 µg /BPA/kg) correlated with insulin hypersecretion, while 50 µg BPA/kg was associated with reduced insulin secretion in both rat offspring and dams (5- and 52-week-old mice) [45].

Susiarjo et al. demonstrated that BPA in male C57BL/6 mice exposed through diet to low doses of BPA (10 µg/kg/d), but not high doses (10 mg/kg/d), developed obesity in adulthood and displayed significantly increased insulin levels [46]. Additional studies have shown that prenatal (GD 10–GD 18) exposure to a low dose of BPA (10 µ/kg /day) subcutaneously resulted in no effect on fasting blood glucose or insulin levels in male offspring [47].

García-Arevalo et al., reported that a subcutaneous injection of a low BPA dose (10 µg/kg/day) in OF-1 mice from GD 9 to GD 16 was associated with increased fasting blood glucose and higher insulin levels at 28 weeks, an increase in the hepatic triglyceride content and increased body weight at postnatal day (PND) 196, and increased BW and weight of fat pad mass [46]. Similarly, Song et al. indicated that the administration of low doses of BPA (1 and 10 µg/mL) in Sprague–Dawley (SD) male rats increased the total BW at PND 7-PND 100. In addition, at puberty, low-dose exposure (10 µg/mL BPA) increased blood glucose and insulin levels and decreased serum ADP [48].

Moreover, a study to investigate the effect of prenatal exposure to BPA altered mouse foetal pancreatic morphology and islet composition; interestingly, they estimated that the maximal level of BPA in their mouse model was 20 ng/mL, which is within the range of levels to which pregnant women are exposed (0.5–22.3 ng/mL). The findings of this study showed that exposure to BPA via maternal diet disrupted pancreatic islet morphology and islet cell composition in foetal mice and increased glucagon in islets [49].

In summary, according to animal model experiments, low levels of BPA prenatal exposure caused alterations in glucose homeostasis, which was reflected in the alterations in fasting glucose levels, insulin release, and secretion, as well as the intolerance of glucose and insulin in childhood. In addition, prenatally BPA-exposed offspring suffered from high glucose levels and insulin resistance in adulthood compared with those in the control groups. Moreover, its effects were sex-specific, as most studies showed impaired glucose homeostasis in male more than in female offspring (Table A1).

#### 6.1.2. High and Low Doses of Developmental BPA Exposure and Lipid Parameters in the Animal Models

The prevalence of overweight and obesity is significantly increasing. In 2020, nine million children under the age of five were either overweight or obese. Obesity is defined by the World Health Organization as abnormal or excessive fat accumulation that may impair health and is classified using body mass index (BMI) [50]. Obesity is related to an increase in circulating triglycerides, FFAs, and leptin, in addition to increased weight and fat accumulation [29].

Recently, a study was conducted to examine the impact of prenatal exposure to a very low dose of BPA (2.5 µg/kg/day) on hepatic lipid metabolism in male and female SD rat foetuses. They assessed the effect of very low BPA exposure on lipid metabolism parameters in pregnant rats. The findings of the study showed no significant differences in serum lipid profiles between BPA-exposed animals and vehicle control animals. In addition, the authors measured the effects of prenatal exposure to 2.5 µg/kg/day of BPA, and the results showed no changes in hepatic cholesterol or triglyceride content in rat foetuses when comparing the exposed and non-exposed groups [44].

In a study conducted in C57BL/6J mice to investigate gestational exposure to 1, 10, 100, and 1000 µg/kg/day of BPA, the study demonstrated that oral exposure to low (1 µg/kg/day) and high (1000 µg/kg/day) doses of BPA significantly increased the hepatic TG content in both male and female offspring. In the same study investigating the effect of gestational exposure to a low dose of BPA (1 µg/kg/day) on hepatic lipid accumulation in 14-week offspring, it was shown that gestational exposure to a low dose of BPA in adult male offspring caused increased hepatic TG and glycogen levels [43].

A recent study conducted by Tonini et al. to examine the impact of prenatal exposure to low-dose BPA (2.5 µg/kg/day) on the livers of SD rat foetuses showed that BPA had no effect on TG or total, LDL, or HDL cholesterol in the serum samples of mother rats, and there was no effect on hepatic TG or total cholesterol content in foetal rats [51]. In contrast, C57BL/6J mice exposed to a low dose of BPA (5 μg/kg/day) showed that low gestational BPA exposure significantly increased BW at 13, 14, 20, and 22 weeks and caused a significant decrease in hepatic TG content. Moreover, in male mice, low gestational BPA exposure significantly decreased BW at 4 weeks and significantly decreased plasma TG, FFA, UC, and VLDL, but not TC (total cholesterol), HDL, or LDL; in addition, these mice showed significantly increased hepatic total cholesterol content [44].

A study in SD rats exposed orally to a high dose of BPA (100 μg/kg/day) showed that the male offspring had significantly increased levels of both hepatic TG and FFA. The findings of this study showed that BPA tended to increase BW at PND 97 and energy intake, although this finding was not statistically significant. There was no significant difference in the fat/lean mass ratios during the birth and weaning periods. However, during the post-weaning to adult period, BPA significantly increased the fat/lean mass ratio (PND 60–90). Furthermore, in females, high BPA showed no effect on BW, cumulative energy intake, fat/lean mass ratio, hepatic TG, or FFAs [52].

Perinatal exposure to 50 µg/kg/day BPA contributes to the obese phenotype and metabolic disorders. In 2017, Malaisé et al., noted that at PND 45, mice offspring perinatally (GD 15–PND 21) exposed to BPA were associated with a decrease in perigonadal white adipose tissue (gWAT). Nevertheless, from PND 70 to PND 170, BPA offspring showed a statistically significant increase in BW compared to vehicle offspring. Although perinatal exposure to BPA increased weight gain in mice, it did not affect food intake. The weight gain of BPA-exposed offspring was associated with an increase in gWAT weight at PND 170, which was not observed at PND 45 [53].

In experimental studies, prenatal BPA exposure has also been shown to disrupt lipid metabolic mechanisms, suggesting that it may increase BW, hepatic TG and FFA, serum TG, as well as total, LDL, and HDL cholesterol in diverse environmental doses, thus contributing to obesity and metabolic disorders. In addition, BPA has sex-specific effects on lipid parameters in animals [46,52,54].

In summary, animal studies have shown that prenatal BPA exposure, even at very low concentrations (<4 μg/kg/day), may decrease birth weight and body weight in children. Moreover, it may cause the dysregulation of the lipid metabolism in offspring and childhood. Moreover, its effect was clear in adulthood, as many studies showed an increase in offspring BW and body fat composition, as well as increased lipid parameters, including hepatic TG and FFA (Table A1).

**Table 2 nutrients-14-02766-t002:** Metabolic parameters observed in animal studies following prenatal exposure to bisphenol A.

Reference	Dose	Low/High Dose	Route of Exposure	Exposure Window	Strain, Species	Sex	Outcomes
[51]	2.5 µg/kg/day	Very low	Drinking water	30 days before mating and continued until GD 20	SD rats	M (male) and F (female)	No effect on parameters of lipid metabolism on pregnant rats. No effect on foetus hepatic TG or TC content.
[43]	1, 10, 100, and 1000 µg/kg/day	Very low, low, and high	Gavage	GD 7.5–GD 16.5	C57BL/6J mice	M and F	In male offspring (14 weeks of age), the hepatic TG content increased significantly at 1 and 1000 µg/kg/day (very low and high doses, respectively) and liver weight increased at very low BPA dose (1 µg/kg/day). In female offspring, the hepatic TG content increased significantly at 1000 µg/kg/day (high dose).
[43]	1 µg/kg/day	Very low	Gavage	GD 7.5–GD 16.5	C57BL/6J mice	M and F	In ♂ (14 weeks) offspring, gestational exposure to low-dose BPA caused increased hepatic TG and glycogen levels and significantly elevated fasting blood glucose and insulin levels, as well as the blood glucose levels, in IPGTT and IPITT analysis. Low-dose gestational exposure to BPA did not impair glucose or lipid metabolism in adult female offspring.
[44]	5 μg/kg/day	Low	Pipette	GD 19–GD 21	C57BL/6J mice	M and F	Low gestational BPA exposure (♀) significantly increased BW (at 13, 14, 20, and 22 weeks), significantly decreased hepatic TG content, and had no effect on plasma lipid parameters, plasma insulin level, or glucose tolerance at 10 and 21 weeks. Low gestational BPA exposure (♂) significantly decreased BW (at 4 weeks), showed faster glucose clearance at 10 and 21 weeks, and significantly decreased plasma TG, FFA, UC, and VLDL, but not TC, HDL, or LDL; there was a significant increase in hepatic TC content.
[52]	100 μg/kg/day	High	Gavage	GD 6–PND 21	SD rats	M and F	High BPA did not significantly alter maternal weight gain or birth weight. ♂ BPA only increased body weight at PND 97 and energy intake (but not significant) and had no effect on body composition (fat/lean mass ratio) at birth or weaning. However, during the post-weaning to adulthood period, BPA significantly increased the fat/lean mass ratio (PND 60–90). ♂ High BPA significantly increased the amount of both hepatic TG and FFAs (PND 1).
[55]	2.5, 25, or 250 µg/Kg/day	Very low, low, and high	Diet	Month virgin state plus 20 days during pregnancy	SD rats	M and F	During pregnancy, the mother’s weight in the third week of pregnancy was significantly lower in the 2.5 µg/Kg/day BPA group. Foetal weight at birth was significantly increased in rats treated with 2.5 µg/kg/day BPA.
[45]	0.5 or 50 µg/Kg/day	Very low and low	Drinking water	GD 3.5–PND 22	Fischer 344 rats	M and F	(♀, ♂) A very low BPA dose (0.5 µg BPA/kg) was correlated with insulin hypersecretion, while 50 µg BPA/kg was associated with reduced insulin secretion from both rat offspring and dams (5 and 52 weeks, respectively).
[56]	0.05, 0.5, or 5 mg/Kg/day	Very low, low, and high	Subcutaneous	GD 30–90	Sheep	F	(♀ at ∼21 months) No effect on plasma TG content for all BPA doses. Significantly increased both hepatic and muscular TG content at 0.5 and 5 mg/Kg/day doses. The plasma LMW ADP levels were significantly reduced only in the BPA high-dose (5 mg/Kg/day) group.
[57]	10 μg/kg/day and 10 mg/kg/day	Low and high	Diet	2 weeks prior to mating (preconception) until weaning	C57BL/6 mice	M and F	♂ Low BPA doses in the male offspring were associated with lower birth weights (PND 1) and the development of obesity in adulthood (PND 98 and117). ♂ Glucose tolerance test results (GTTs) were greater in high-dose (10 mg/kg/day) BPA male offspring than in control male offspring, consistent with the phenotype of glucose intolerance (PND 98 and 117). ♂ significantly increased insulin levels in the low-dose BPA group (PND 98 and 117). ♀ and ♂ Low and high doses of BPA had no effect on fasting glucose levels. Low- and high-dose BPA-exposed male but not female offspring had impaired glucose homeostasis and developed obesity in adulthood.
[53]	50 µg/kg/day	Low	Oral gavage	GD 15–PND 21	C3H/HeN mice	M	Low doses decreased gWAT at PND 45 and increased BW (PND 70 until PND 170). However, there was no effect on food intake, and it was associated with glucose intolerance (at PND 35) and decreased insulin sensitivity (at PND 125).
[54]	0.5 or 50 µg/kg/day	Very low and low	Drinking water	GD 3.5–PND 22	F344 rats	M and F	No effect on maternal parameters, including BW, food, and water intake. Increase in plasma TG with 50 µg/kg/day (♀) at 5 weeks and slightly increased BW at 52 weeks. Increase in plasma TG at 5 weeks in ♂ with 0.5 µg/kg BW/day offspring and low BW at 52 weeks. No effect on ♂ or ♀ offspring parameters, plasma TG, cholesterol, leptin, or ADP in BPA-exposed 52-week rat offspring. 5-week-old ♂ exposed to low doses (0.5 µg/kg /day) were linked with insulin resistance.
[46]	10 μg/kg/day	Low	Subcutaneous injection	GD 9–GD 16	OF-1 mice	M	Increased fasting blood glucose and higher insulin level (28 weeks), increased hepatic TG levels, and increased BW (PND 196). Increase in perigonadal fat pad weight at 28 weeks.
[48]	1 and 10 µg/mL	Low	Drinking water	GD 6–PND 21	SD rats	M	1 µg/mL or 10 µg/mL BPA in water had no significant effect on maternal weight gain, food intake, or water consumption during pregnancy. For ♂ PND 7–PND 100, low-dose BPA (1 and 10 µg/mL) exposure increased BW. ♂ At puberty stage, a low dose (10 µg/mL BPA) increased blood glucose and insulin and decreased serum ADP. ♂At adult stage, both low doses increased blood glucose and insulin and significantly decreased the level of serum ADP.
[58]	50 μg/kg/day	Low	Oral gavage	GD 0 until weaning at 3 weeks	Wistar rats	M	At week 3, a low dose had no effect on BW or blood glucose but increased the serum insulin level. At week 21, a low dose increased BW and insulin resistance and impaired glucose tolerance in rat’s hepatic tissue.
[26]	1 and 20 µg/kg diet BPA	Very low and low	Diet	Pregnant and lactating	CD-1 mice	M and F	Both low doses of BPA had no effect on maternal body weight or food intake or on the BW of 3-month-old (♂ and ♀) offspring. ♂ For adults, a low dose (20 µg/kg) impaired glucose tolerance and increased plasma insulin. However, no effect on leptin level was found. ♀ No effect on glucose tolerance and increased plasma leptin.
[59]	10 or 100 µg/kg/day	Low and high	Subcutaneous	GD 9–GD 16	OF-1 mice	M	A low BPA dose (10 µg/kg/day) displayed maternal glucose intolerance, and a high dose of BPA (100 µg/kg/day) showed a tendency to increase insulin sensitivity (but not statistically significant). Pregnant mice at GD 18 with exposure to BPA (100 µg/kg/day) showed significantly increased leptin, TG, and glycerol. At 3 months of age, for ♂ and ♀, both BPA doses had no effect on insulin resistance or glucose intolerance. At 6 months of age, male offspring exposed in utero had reduced glucose tolerance and increased insulin resistance (10 µg/kg/day). A low BPA (10µg/kg/day) dose decreased BW in the offspring. BPA (100 µg/kg/day) mice had higher levels of leptin than F0-BPA 10µg/kg/day mice.
[49]	25 mg BPA/kg diet	High	Diet	GD 7.5–GD 18.5	C57BL/6 mice	Foetus	A high dose of BPA (25 mg/kg) did not alter the expression of insulin in the foetal pancreatic islets. However, there was an increase in glucagon expression in BPA-exposed foetal pancreatic islets.
[60]	100 µg/kg/day	High	Subcutaneous	GD 6–PND 0 PND 0–PND 21 GD 6–PND 21	C57BL6 mice	M and F	No effect on birth weight. ♂ and ♀ Showed significant decreases in the body weight of pups in the GD 6–PND 0 group. ♀ No differences in body weight were observed among the GD 6–PND 0, PND 0–PND 21, or GD 6–PND 21 groups from 3 to 35 weeks. However, GD 6–PND 0 mice started gaining less weight than controls from weaning until 35 weeks. ♂ The GD 6–PND 0 and PND 0–PND 21 groups increased BW, but the GD 6–PND 0 group had deceased BW. ♀ No effect on glucose tolerance in the PND 0–PND 21 or GD 6–PND 21 groups from 3 months to 8 months. PND 0–PND 21 male mice showed glucose intolerance.
[61]	10 μg/kg/day and 10 mg/kg/day	Low and high	Diet	2 weeks prior to mating until weaning	C57BL/6J mice	M and F	High and low doses of BPA significantly impaired insulin secretion in male but not in female offspring.
[62]	50 ng, 50 µg, or 50 mg BPA/kg	Very low, low, and high	Diet	2 weeks prior to mating, pregnancy, and lactation	Mice	M and F	♀ 50 ng and 50 mg of BPA/kg in the diet increased serum ADP levels in female adult offspring. ♂ Males exposed to 50 µg BPA/kg had marginally significant lower levels of ADP. ♀ 50 mg of BPA/kg in the diet decreased the mean baseline glucose and insulin levels. ♀ Significant decrease in insulin and the homeostasis assessment model of insulin resistance (HOMA-IR) levels but mildly lower serum glucose levels with 50 mg/kg in the diet in the female offspring. ♀ Significantly decreased serum leptin (50 mg BPA) but no effect in ♂. ♀ At 6 and 9 months of age, 50 ng BPA/kg-exposed females had an increase in fat mass and body weight compared with controls. ♂ No significant changes were found in fat mass in males.
[63]	50 µg/kg/day	Low	Drinking water	GD 9–PND 21	Wistar rats	M	Exposure to BPA (50 µg/kg/d) significantly increased BW with significantly greater amounts of epididymal and perirenal fat pads and increased food intake. BPA alone had no effect on fasting glucose or the glucose tolerance test, but BPA exposure of 50 µg/kg/day increased insulin and leptin levels. BPA impaired glucose homeostasis, induced obesity, and increased food intake in adult male rats.
[64]	0.1 mg/L	Low	Drinking water	GD 11–PND 21	SD rats	M	A low BPA dose had no effect on the food intake or water consumption of dams. BPA had no effect on BW of either dams or pups. A low BPA dose led to decreased glucose metabolism in rats.
[65]	0.05, 0.5, or 5 mg/kg/day	Very low, low, and high	Subcutaneous injections	GD 30–GD 90 147 days	Sheep	F	Prenatal-BPA-treated adult females were hyperglycaemic and had IR only at low BPA doses.
[66]	2.5, 25, and 250 µg/kg/day	Very low, low, and high	Drinking water	Month virgin state plus 20 days during pregnancy	SD rats	M and F	BPA had no effect on the weight of pregnant rats or their plasma lipid profile (TC, TG, LDL, and HDL); sacrificed at GD 20.
[67]	20 and 40 µg/kg/day	Low	Orally by gastric intubation	GD 1 to GD 20	Rats	Foetus	Increase in foetal serum leptin and insulin levels in both treated groups compared to controls. However, there was a decrease in maternal and foetal body weight in both treated groups compared to the control group.

Male (♂); female (♀); triglyceride (TG); gestational day (GD); postnatal day (PND); bisphenol A (BPA); unesterified cholesterol (UC); very low-density lipoprotein (VLDL); total cholesterol (TC); high-density lipoprotein (HDL); low-density lipoprotein (LDL) was intraperitoneal glucose tolerance test (IPGTT); intraperitoneal insulin tolerance test (IPITT); body weight (BW); free fatty acid (FFA); adiponectin (ADP); and perigonadal white adipose tissue (gWAT).

### 6.2. Metabolic Effect of Prenatal BPA Exposure in Children (Epidemiological Studies)

In the literature, studies have shown conflicting findings on the outcomes of BPA exposure. We found six epidemiological studies that investigated the effect of prenatal BPA exposure on metabolic parameters in children, as summarised in Table 3.

A recent cohort study in Taiwan conducted on 162 mother–child pairs showed a correlation between BPA exposure in the second trimester and low birth weight. In this study, the limit of detection (LOD) for BPA in urine was 0.16 ng/mL. Urine BPA concentrations above the 75th percentile were classified as having a high BPA range (12.76–114.95 µg/g creatinine), and there was also a group with a low BPA range (0.01–8.37 µg/g creatinine) in all three trimesters. However, they did not mention prenatal exposure to BPA at which the dose caused low birth weight in newborns [68].

A prospective cohort study was conducted at a maternity and child health hospital in Shanghai, China. The results showed that a moderate maternal prenatal BPA level (1.14 ng/mL) was associated with higher plasma glucose levels in boys. In contrast, in girls, the plasma glucose level was lower (0.26 mmol/L) with moderate prenatal BPA levels; however, the difference was not significant. The study also showed no associations between prenatal BPA exposure and children’s body weight, BMI, skinfold thickness, serum lipid levels, or insulin levels in children aged two years old, either for girls or boys [69].

In 2017, a cross-sectional study was conducted on 250 Mexican mother–child pairs to investigate the relationship between prenatal and childhood exposure to BPA and phthalates on BMI z-score, waist circumference, and the sum of tricep and subscapular skinfold thicknesses in Mexican children. Spot urine was collected from mothers in the third trimester and at 8–14 years old from children. The findings showed that increased BPA exposure was positively associated with the sum of skinfold thicknesses and BMI z-scores in girls but not in boys [70].

Vafeiadi et al. analysed BPA levels in spot urine samples collected from pregnant Greek women in the first trimester of pregnancy and their children at two-and-a-half and four years of age using an Olympus 2700 immunoassay system. In this study, the LOD was equal to 0.01 ng/mL, and the average %>LOD was >99% in mothers and children. This study showed that urinary BPA concentrations were lower in pregnant women than in their children. Higher prenatal BPA concentrations were associated with BMI and adiposity measures that were lower in girls and higher in boys aged 1–4 years. However, there was no substantial evidence that BMI was different among children with high prenatal BPA concentrations compared to those with low prenatal BPA concentrations, based on the 80th percentile [71].

Notably, the association between prenatal exposure to BPA and the function of metabolic markers appeared to be modified by sex. Martin et al. found an inverse relationship between maternal urinary BPA and ADP levels in men. In contrast, female infants were shown to have higher leptin levels than males [72].

A birth cohort study conducted on 537 mother–child pairs of Mexican-American origin (LOD = 0.4ng/mL) showed that late-pregnancy urinary BPA levels (26 weeks) were associated with increased leptin levels in boys, while early-pregnancy BPA levels (13 weeks) were positively related to ADP levels in 9-year-old girls [72].

Remarkably, these cohort studies determined the relationship between prenatal exposure to BPA and its effect on metabolism in children. However, most of these studies were not designed to investigate the relationship between prenatal BPA exposure and biochemical parameters. Some biochemical parameters were determined, such as birth weight, waist circumference, lipid profile, glucose and insulin levels, and adipokine levels. Briefly, the results of these studies showed that prenatal BPA exposure affected the metabolic parameters of children in some manner. Nevertheless, exposure to BPA during pregnancy differed in its effect on children in terms of whether the exposure period was during the early or late period of pregnancy, as well as in terms of sex.

**Table 3 nutrients-14-02766-t003:** Metabolic parameters observed in human studies following prenatal exposure to bisphenol A.

Author (Year) Reference	Country	Study Design (N)	LOD	%>LOD	BPA Assessment	Collection Year(s)	Outcome Time Window	Results
Huang Y et al., (2021) [68]	Taiwan	Cohort study: 162 mother–infant pairs	0.16 ng/mL	-	Ultra-performance liquid chromatography coupled with time-of-flight mass spectrometry	2010	Mothers: three spot urine samples and three blood samples at approximately 11 and 26 weeks gestation and at admission for delivery Children: cord blood samples	Exposure to BPA in the second trimester was associated with low birth weight.
Ouyang F et al., (2020) [69]	China	Birth cohort study: 218 pregnant women	0.1 µg/L = (0.1ng/mL)	Mothers = 98.2% children = 99.4%	HPLC-MS/MS	2012–2013	Spot urine samples Mothers: third trimester Children: 2 years	In boys, a medium maternal prenatal BPA level (1.14 µg/L) was associated with higher plasma glucose. No associations were found between prenatal BPA and child body weight, BMI, skinfold thicknesses, serum lipids, or insulin in children, either girls or boys.
Yang T et al., (2017) [70]	Mexico	Cross-sectional: 250 mother–child pairs	0.4 ng/mL	Prenatal = 70% Children = 85%	Isotope dilution-liquid chromatography-tandem mass spectrometry	2012	A single spot urine sample for both Mothers: third trimester Children: 8–14 years	Increased BPA exposure was positively associated with the sum of skinfold thicknesses and BMI z-score in girls but not in boys.
Vafeiadi M et al., (2016) [71]	Greece	Cohort study: 500 mother–child pairs	0.01 ng/mL	Mothers = 99.6% Children at 2.5 years = 99.6% Children at 4 years = 98.8%	OLYMPUS 2700 immunoassay system	2007	A spot urine sample for mothers in the first trimester of pregnancy Children: two spot urine samples at 2.5 and 4 years of age	Prenatal BPA concentrations were associated with lower BMI and adiposity measures in girls and higher measures in boys at 1–4 years of age. Maternal BPA was not significantly associated with birth weight.
Martin J et al., (2014) [72]	Canada	Cohort study: 2001 women	0.2ng/mL	86.6%	GC-MS-MS instrument with a GC Agilent 6890 N (Agilent Technologies; Mississauga, ON, Canada) coupled with a tandem mass spectrometer Quattro Micro GC	2008–2011	Urine samples Mothers: first trimester	In males, there was an inverse relationship between maternal urinary BPA and ADP levels. Female infants were shown to have higher leptin levels than male infants.
Volberg V (2013) [73]	Mexico/USA	Birth cohort study: 537 mother–child pairs	5 0.4 µg/L = 0.4ng/mL	Mothers at 13-week gestation = 79% and mothers at 26-week gestation = 83% Children = 91%	Online solid-phase extraction coupled with isotope-dilution high-performance liquid chromatography tandem mass spectrometry with peak focusing	1999–2000	Two urine spot samples Mothers at 13 and 26 weeks gestation Children at 9 years	No differences were observed comparing maternal and child anthropometric measures, including BMI, food consumption, and child birth size. Late-pregnancy urinary BPA levels were associated with increased leptin levels in boys, while early-pregnancy BPA levels were positively related to ADP levels in 9-year-old girls.

### 6.3. Why do Low Doses Have More Effect Than High Doses?

According to Welshons et al., (2003), the different responses between low and high doses of BPA in different studies are due to a common mechanism by which BPA affects the metabolic pathway. BPA is able to impair hormonal signalling in the endocrine system via direct interaction with oestrogen receptors alpha or beta (ER-α and β, respectively) present in the cell. Thus, the mechanism is receptor occupancy, which demonstrated that the low doses altered the hormone concentration in a manner that may be sufficient to induce an oestrogenic hormonal response (bioactive concentration), more so than high doses [74].

Another predicted mechanism is the non-genomic signalling pathway, which includes GPR30. As a result, low-dose BPA stimulates the GPR30 oestrogen receptor and induces the production of proinflammatory cytokines from adipocytes. Among numerous cytokines, IL-8 has been associated with chronic inflammatory processes in subjects with severe obesity, T2DM, atherosclerosis, cardiovascular disease, and cancer [41].

## 7. Conclusions

In this review, we investigated the effect of prenatal exposure to a wide range of BPA doses on metabolic parameters in the offspring. Firstly, we confirmed that BPA exposure had a sex-specific effect. Epidemiological and animal studies have demonstrated that developmental exposure to BPA may lead to impaired glucose and lipid homeostasis, obesity, and increased food intake in male offspring. Moreover, studies have also shown obesity and insulin resistance in adult men due to BPA exposure. The findings of our review also showed that exposure to low BPA doses (<4 µg/kg/day) during critical periods early in life caused the disruption of glucose and lipid metabolism, and more weight gain was induced in the groups exposed to low doses prenatally than those exposed to moderate or high doses, especially in male offspring. Exposure to BPA during the perinatal period causes changes in metabolic parameters in childhood and increases the risk of metabolic diseases, such as obesity and T2DM, in adulthood.

Finally, in view of this, future studies are needed in humans and animals to examine the biochemical parameters to understand the effect of maternal exposure to different BPA doses during pregnancy and its effect on metabolic parameters in children.

## Figures and Tables

**Figure 1 nutrients-14-02766-f001:**
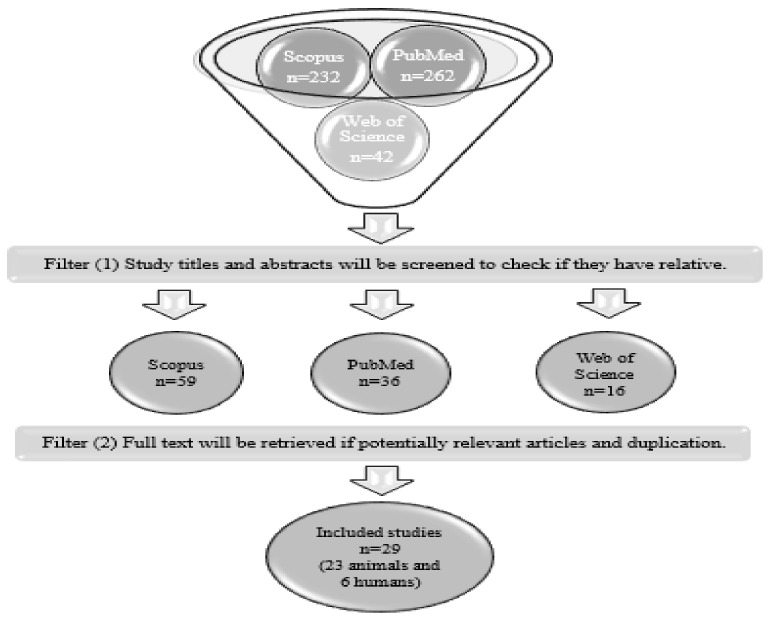
Study search strategy.

**Figure 2 nutrients-14-02766-f002:**
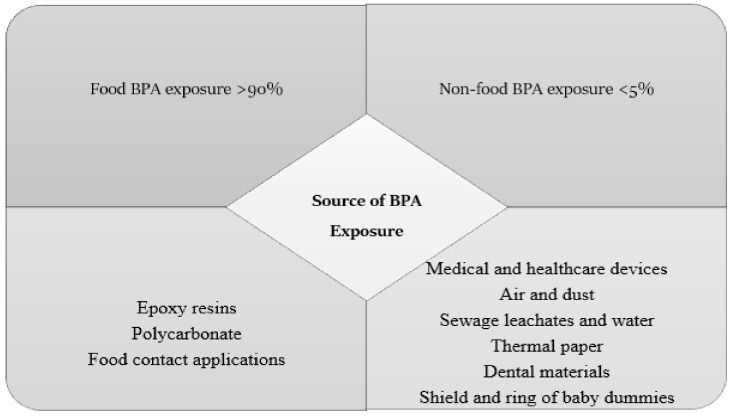
Food and non-food exposure to BPA sources.

**Table 1 nutrients-14-02766-t001:** Patient, intervention, comparator, and outcome (PICO) criteria for inclusion and exclusion.

	Included Criteria	Excluded Criteria
Population	Experiment in animals and humans	All non-human and non-animal studies
Intervention	Prenatal BPA exposure	Non-BPA exposure in mixed chemical treatments, transgenerational studies, or direct exposure
Comparator	Low and high doses relative to control	
Outcome	Metabolic syndrome, glucose homeostasis distribution, lipid homeostasis distribution, diabetes mellitus, disrupted insulin release and level	All other non-metabolic complications or no determination of metabolic parameter outcomes

## Data Availability

Not applicable.

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
