# Peer review of "The Current Findings on the Impact of Prenatal BPA Exposure on Metabolic Parameters: In Vivo and Epidemiological Evidence"

_nutrients, 2022, doi:10.3390/nu14132766_

Round 1
Reviewer 1 Report
Why the Authors limited themselves to reviewing publications from the last fifteen years> S0ome aspects connected with correlation between exposure to BPA with metabolic syndrome has been described before 2006. He review of such studies would increase the value of the manuscript.
A diagram of the sources of bisphenol from which it can affect living organisms ( with products, in which BPA is present in the greatest amount, added to the introduction would increase the attractiveness of the manuscript
Some information about metabolism of BPA in human organisms could be added into the manuscript
Conclusion is not very readable. I suggest editing the conclusion to make it more concise and clear.
Author Response
We thank the editorial office of Nutrition and Metabolism of Nutrients Journal and the reviewers for their most insightful comments and invaluable input. We have made the suggested changes. (C stands for comment, R stands for response). All modifications to our manuscript are in blue and red fonts.
We hope that the revised manuscript meets these expert reviewers’ expectations.
Reviewer 1
C1: Why the Authors limited themselves to reviewing publications from the last fifteen years> S0ome aspects connected with correlation between exposure to BPA with metabolic syndrome has been described before 2006. He review of such studies would increase the value of the manuscript.
R1: Thank you for the comment. We decided to start the review from 2006 as the European Food Safety Authority (EFSA) had completed its first full risk assessment of bisphenol A in 2006. Therefore, we believe that there are more updated studies by experts and researchers based on this EFSA report (2006).
C2: A diagram of the sources of bisphenol from which it can affect living organisms (with products, in which BPA is present in the greatest amount, added to the introduction would increase the attractiveness of the manuscript
R2: Thank you for the comment and suggestion. Figure 2 is already included to show the sources of BPA. However, a paragraph is also added to add more information on this matter (Pages 3-4, lines117-119).
C3: Some information about metabolism of BPA in human organisms could be added into the manuscript
R3: Thank you for the comment and suggestion, the paragraphs about BPA metabolism are added (page 4, lines 128-142).
C4: Conclusion is not very readable. I suggest editing the conclusion to make it more concise and clear.
R4: Thank you for the comment and suggestion. The conclusion is edited accordingly (page29, lines 389-421)
Reviewer 2 Report
I think it is a great review. It has been demonstrate a deep knowledge about the topic. I really apprecciate the way of explaining this difficutl topic.
Author Response
We thank the editorial office of Nutrition and Metabolism of Nutrients Journal and the reviewers for their most insightful comments and invaluable input. We have made the suggested changes. (C stands for comment, R stands for response). All modifications to our manuscript are in blue and red fonts.
We hope that the revised manuscript meets these expert reviewers’ expectations.
C1: I think it is a great review. It has been demonstrate a deep knowledge about the topic. I really appreciate the way of explaining this difficult topic.
R1: Thank you for the comment. We really appreciate your valuable opinion.
